# Research on Coupled Cooperative Operation of Medium- and Long-Term and Spot Electricity Transaction for Multi-Energy System: A Case Study in China

**Kaiyan Wang [1,2], Xueyan Wang [1,2,*], Rong Jia [1,2], Jian Dang [1,2], Yan Liang [1,2] and Haodong Du [1,2]**

1    College of Electrical Engineering, Xi'an University of Technology, Xi'an 710048, China
2    Shaanxi Union Research Center of Xi'an University of Technology and Enterprise for Electrical Power & Integrated Energy, Xi'an University of Technology, Xi'an 710048, China
*    Correspondence: 3170651062@stu.xaut.edu.cn

**Abstract:** Due to the intermittent and anti-peak shaving characteristics of the new energy generator sets, the phenomenon of power abandonment hinders direct participation in the electricity market transactions. The hybrid electricity market can use spot market transactions to absorb renewable energy to a large extent. The multi-energy complementary operation coupling of the hybrid electricity market transactions can exploit the complementation and substitution between different energy sources, realize flexible energy production, consumption, storage, and transmission, and optimize the allocation of resources on a larger scale. In this paper, a mid-long-term spot transaction coordination scheduling (MTCS) model for a multi-energy system is constructed by considering the medium- and long-term electricity market uncertainty and the trial operation characteristics of the spot power market in China. A two-stage solution method is introduced to solve the complex multi-agent, multi-period, and multi-energy model. The results of testing this model on the Gansu region, one of the first eight spot pilot areas in China, are presented and discussed in detail. The results showed that this MTCS model could reduce the opening of thermal power units to a more considerable extent, prioritize the consumption of new energy power generation, and reduce the output uncertainty of new energy through the hybrid power market.

**Keywords:** electricity spot market; medium- and long-term transaction; renewable energy; multi-energy complementary system; coupled cooperative operation

## 1. Introduction

The low-carbon transformation and development of the energy system is a standard solution for the international community to deal with climate, environmental, and energy issues [1]. The benefits of coordinated operation and comprehensive energy system optimization have received extensive attention [2]. The combined process of the coupled trading of multi-energy systems and multi-energy markets allows flexible production, consumption, storage, and transmission of energy by exploiting the complementarity and substitutability between different energy sources. It optimizes resource allocation on a larger scale and absorbs renewable energy sources to promote the low-carbon transformation and development of the energy systems [3]. In the above context, renewable energy is increasingly introduced into the energy system and participates in electricity market transactions. However, new energy power generation is significantly affected by weather changes. It has the characteristics of volatility, randomness, and intermittent news, which hinders the enthusiasm for its participation in the power market. Therefore, coordinated multi-energy system power market transactions can effectively improve renewable energy's supply and demand coordination capabilities, promote clean energy production and nearby consumption, and incorporate it into the unified national power market system, which is an inevitable trend for future development.

The power industry is the primary source of China's carbon emissions. Promoting the transformation of energy and power and building a power system based on new energy are the internal mission of power companies and the external demand under the unique situation. Among them, the market-oriented electricity reform is a crucial area to be promoted. Under increasingly severe global climate change issues and the goal and vision of achieving carbon peaks, the development of renewable energy and the promotion of energy conservation and emission reduction have become essential tasks for developing electricity in various countries [4]. China's electric energy market has been discussed frequently. As the effective use of renewable energy is limited by the traditional institutions and practices of China's power industry, a new example of the structural innovation and reform of the power industry is needed to overcome the challenges of renewable energy (RE) development [5]. The electricity spot market can promote the formation of a clean, safe, low-carbon, and efficient energy system, accelerate the development and consumption of new energy, and ensure the utilization rate of new energy. Hence, efficient cooperative operation of the hybrid power market and a relevant mathematical theoretical model are needed to accelerate the building of RE-friendly electricity spot markets [6]. Northern Europe, Australia, Germany, the United States, the United Kingdom, and other countries started the reform of their power systems early. The power market system is complete, and many types of power transactions exist. Their power market construction aims to optimize resource allocation and achieve a low-carbon energy transition. The Department of Energy and Climate Change of the United Kingdom officially issued a white paper on power market reforms, opening a new round focusing on promoting low-carbon development and ensuring supply security [7]. The United States announced a standard electricity market, mainly focusing on the fairness and openness of the transmission grid, and California has gradually shifted from decentralized to centralized transactions [8]. The Japanese government has passed a new round of power reform plans, proposing reforms such as comprehensively liberalizing competition in power sales and establishing a national dispatch coordination agency and power grids and power generation [9].

Solving the collaborative optimization problem of multi-energy coupled systems has become an essential difficulty in its development. Zhang et al. proposed a multi-stage robust optimization model for the coordinated operation of an electricity-gas-transportation coupled system, which simultaneously considered the uncertainties of traffic demands, wind power, and gas fuel consumption by gas-fired units [10]. Ma et al. concentrated on the wind-hydrogen-heat multi-agent energy system's cooperative planning and operation problems. A coordinated planning and operative model for the wind-hydrogen-heating multi-agent energy system is proposed based on the Nash bargaining game theory [11]. Zhao et al. established a modeling framework for a multi-energy system (MES) with a coordinated supply of combined cooling, heating, and power using solar energy and indicated that the strategy of FOC has the advantage of saving operation costs [12].

Li et al. proposed an intraday, multi-objective, hierarchical, and coordinated operation scheduling method for a multi-energy system (MES) to study energy system participation in market transactions at various time scales and the impact of source and load uncertainties in order to improve energy management [13]. Cai et al. designed a hydro-dominated provincial power spot market mechanism. They analyzed the advantages of a centralized market model for operational security, the optimization of hydropower allocation, and the connection with existing dispatching systems [14]. Wang et al. proposed the inter-provincial power spot market model for the national unified power market and designed the market-clearing algorithm [15]. Mu et al. introduced the present situation of the Yunnan electricity market, including the power structure, monthly transaction price, and transaction modes. They proposed a coordination mechanism between the spot and the forward markets [16]. Goudarzi et al. established an optimal day-ahead power market model using an optimized framework to integrate and manage related uncertain resources to obtain reasonable profits [17]. Bhatia et al. analyzed and evaluated the impact of renewable resources on price forecasts and proposed a generalized architecture of the bootstrap aggregation stack to

encourage market participants to formulate real-time operating strategies [18]. Sahoo and Hota considered the intermittent nature of renewable energy and its many uncertainties. They used an improved whale optimization algorithm to establish a bidding strategy model to maximize the profits of power suppliers [19]. Flammini et al. simulated the future wholesale price of electricity by considering the hourly power generation quotation dataset and using clean, renewable energy to meet the future electricity demand [20]. Dye et al. analyzed the value of point-to-point transactions versus the absence of local markets and the impact of PV, battery, and EV deployment [21]. Simona et al. investigate whether such a renewable energy increase has affected the contagion behavior in the Italian electricity spot market, considering the difference between interdependence and contagion and the direction of the shock [22].

Promoting the construction of a green, low-carbon, and clean energy system with renewable energy as the mainstay is a strategic choice for China and most countries worldwide. However, due to the random and intermittent effects of renewable energy, China's renewable energy consumption problem has become prominent, and it is difficult to get out of the predicament of renewable energy consumption only by tapping the internal potential of the existing power system. A reasonable market mechanism is one of the critical elements in promoting the consumption of renewable energy. Previous research mainly focuses on the independent operation of medium- and long-term transactions or the conceptual level of spot transactions, which lacks theoretical guidance for coordinating medium- and long-term trades in spot transactions after multi-energy participation [23–25]. Few models have been designed. Notably, the large amount of new energy participating in market transactions may lead to wind and light abandonment, uneven source and load distribution among regions, and a conflict of interest among market entities. For multi-energy systems, the coupling methods of different energy markets also affect the operation of multi-energy systems and renewable energy consumption. Therefore, in developing multi-energy systems, how to coordinate in an orderly way the electricity determined by the direct transaction and the electricity of the new energy units is the focus of the market construction.

Therefore, as the advance of previous research, this paper uncovers the current issues of multi-energy systems directly participating in the power market and then proposes a mid-long-term spot transaction coordination scheduling (MTCS) model by considering the long-term uncertainty in the hybrid electricity markets. The main contributions of this research are as follows:

(1) This research realizes the medium- and long-term electricity markets and the spot market to be coupled with a cooperative operation for a multi-energy hybrid system.
(2) A new dispatching model to promote fresh energy consumption is proposed to deal with the uncertainty of power decomposition in medium- and long-term contracts and the incompleteness of the spot market pilot operation.
(3) The MTCS model, based on the objective function of minimizing the operating cost of thermal power, can effectively dispatch thermal power and hydropower units to cut peaks and fill valleys and maximize renewable energy consumption.
(4) A two-stage solution method that includes electricity decomposition and unit start and stop status is introduced to solve the complex multi-agent, multi-period, and multi-energy model.
(5) Gansu province is China's earliest pilot spot market region, and typical scenes are introduced to cross-validate the MTCS model.

This paper comprises five sections. The problem statement and study area description will be stated in Section 2. The process of mathematical modeling will be introduced in Section 3. Section 4 will present the calculations, results, and analysis. Section 5 presents a discussion, including a comparison of relevant scenes. This paper will provide a comprehensive conclusion and direction for future research in Section 6.

## 2. Problem Statement and Study Area Description

### 2.1. Study Area

Renewable energy is often characterized by intermittent randomness and volatility, and its rapid development has given rise to the demand for a synergistic multi-energy system power market [26]. Forming the spot trading mechanism helps remove the obstacles to developing large-scale renewable energy. Specifically, energy power generation companies, such as wind power and PV power, can achieve a more significant market by the natural characteristics of the lower marginal cost advantages. Therefore, the spot market provides a possible way to solve the large-scale consumption of new energy power, which can further implement the national power system reform and strategic energy transformation [27].

In 2019, the National Development and Reform Commission and the Energy Administration jointly issued a document to select eight regions to carry out the pilot construction of the electricity spot market. The geographical location of China's spot pilot is shown in Figure 1. Gansu Province is one of China's first spot market pilot provinces and is one of the representative provinces of the Northwest area, which is a rapidly developing area for the wind power and solar power generation industries. However, as the power market reform continues to deepen, Gansu Province in China is facing more and more severe abandonment of wind and solar power.

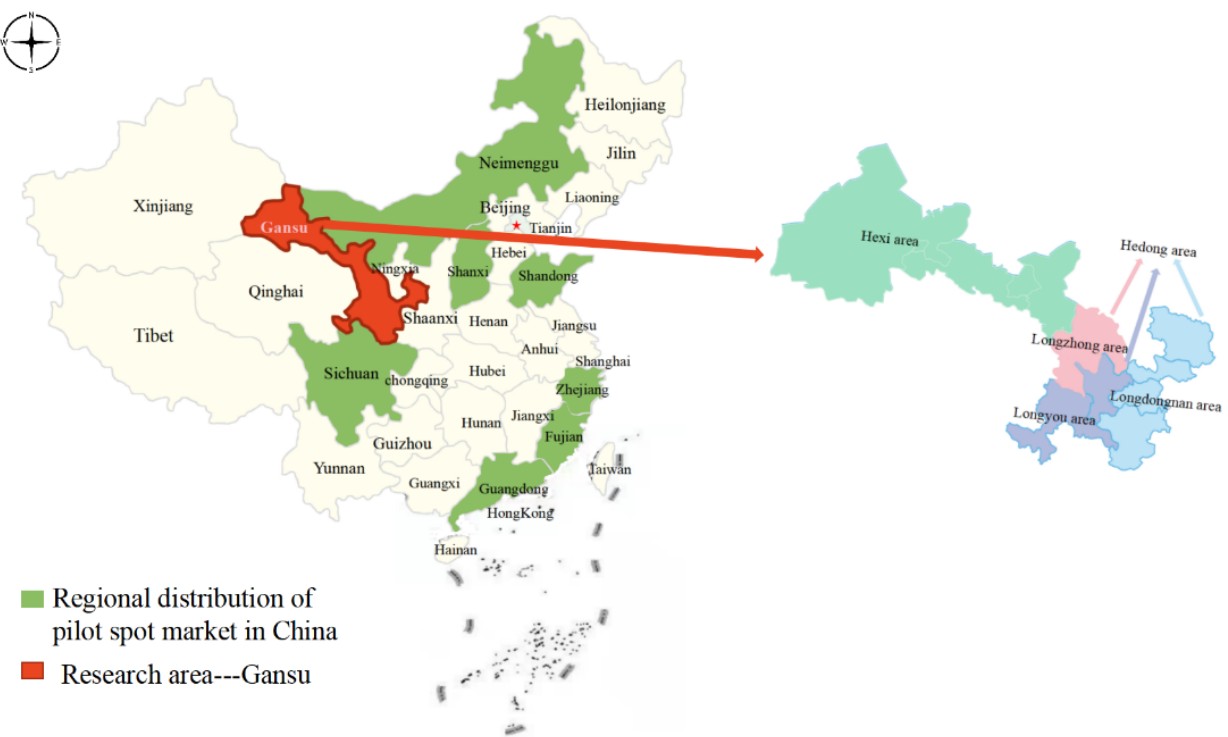

**Figure 1.** Area diagram.

### 2.2. Data Source

The rapid expansion of renewable energy, such as wind and PV power, has become the primary energy in Gansu. In 2020, the installed capacity of wind power in Gansu Province reached 13.121 million kW, and the installed capacity of PV power reached 9,251,100 kW. The installed capacity of new energy accounts for nearly 60% of the total installed power capacity (Figure 2). The power generation capacity of new energy will account for more than 60% of the total electricity consumption of the whole province.

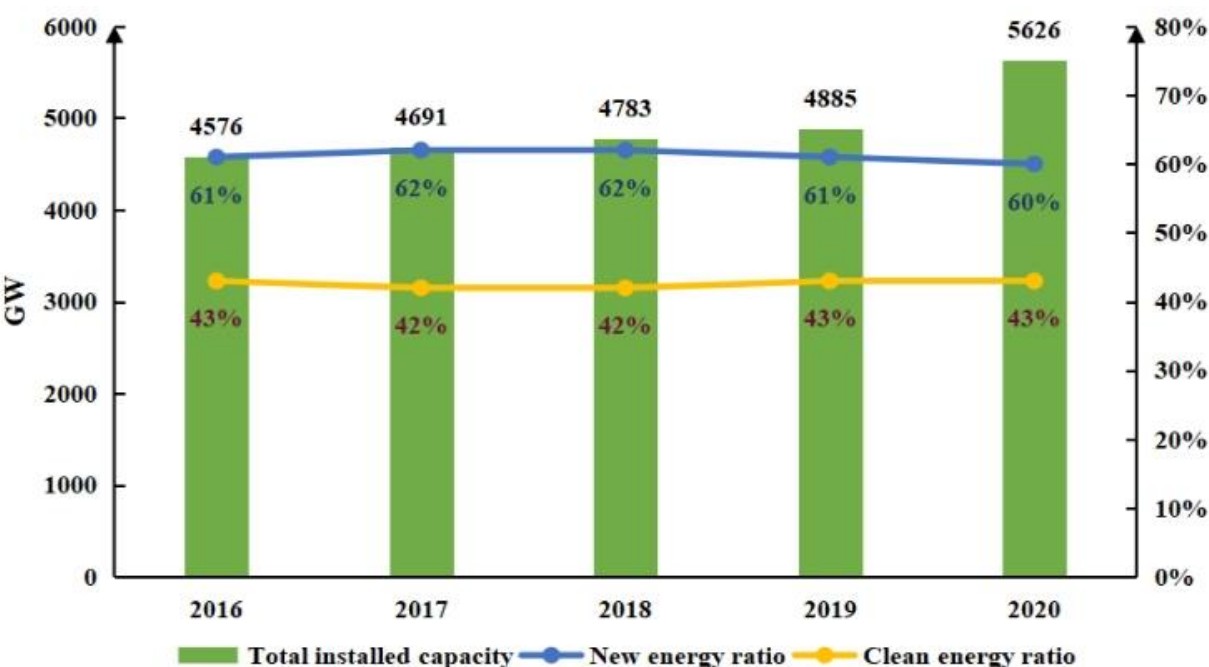

**Figure 2.** Gansu province has a high proportion of new energy production.

*2.3. Problem Summary*

Gansu Province is one of China's first spot market pilot provinces and is one of the representative provinces of the Northwest area. In the past ten years, the installed energy capacity in Northwest China has increased by more than 30 times. Its ability, installed capacity, and new energy penetration rate have ranked first in all the regions of the country for many years. There has been an explosive growth of new energy in the Northwest Power Grid and the grid equivalent load curve. The shift to the "duck shape" has caused profound changes in the demand for peak shaving. The capacity of the power grid peak is essential in the work for the development of new energy. The Northwest Power Grid peak shaving is facing more significant challenges at this stage.

The installed capacity of new energy in Gansu Province has been increasing yearly. Due to the limited local consumption space, cross-regional transportation channels, and insufficient peak shaving capacity, Gansu is also one of the regions with the most severe power abandonment phenomenon [28,29]. Therefore, it has become increasingly difficult to absorb a high proportion of new energy. Gansu Province, one of China's first spot market pilot provinces, is actively exploring a spot market system that promotes new energy consumption and has established a spot market mechanism involving new energy [30]. However, new energy power generation is affected by weather changes, and its volatility, randomness, and other characteristics determine that new energy cannot sign medium- and long-term contracts with curves in the annual and monthly transactions. In the context of the centralized market, the operation of the spot market also requires the settlement curve decomposition of medium- and long-term new energy contracts to connect with the spot market effectively.

## 3. The Development of Mid-Long-Term Spot Transaction Coordination Scheduling Model (MTCS)

*3.1. Basic Framework of the MTCS Model*

The key to improving the ability to absorb renewable energy is to improve the efficiency of the flexible resource allocation and to use a suitable scheduling transaction mechanism to make flexible resources efficiently compensate for the uncertainty of renewable energy. Optimizing the planning and operation of the multi-energy system by using the advantages of the market to allocate resources will bring revolutionary changes to

the traditional energy system [31]. Coordinated operation of the medium- and long-term electricity and spot markets can promote the development and consumption of new energy and ensure the utilization rate of new energy [32]. Specifically, medium- and long-term transactions are conducive to ensuring the power system's safe operation and the electricity market's stable operation. The contracts for medium- and long-term electricity can maintain the high flexibility of the new energy medium- and long-term contract power and guide the hybrid energy system in the spot market. The spot market provides a critical way to settle the matter of large-scale consumption of new energy power [33–37]. The competitive transactions for the spot market can improve the electricity market through efficient unit combination operation. Based on the joint operation mechanism and the spot implementation status, the mid-long-term spot transaction coordination scheduling model (MTCS) is proposed in this paper to maximize the consumption of renewable energy efficaciously. This MTCS model consists of three sections: the mid-long-term spot transaction coordination scheduling model, the two-stage solving process, and the confirmed cases experiment. The basic framework of the MTCS model is shown in Figure 3.

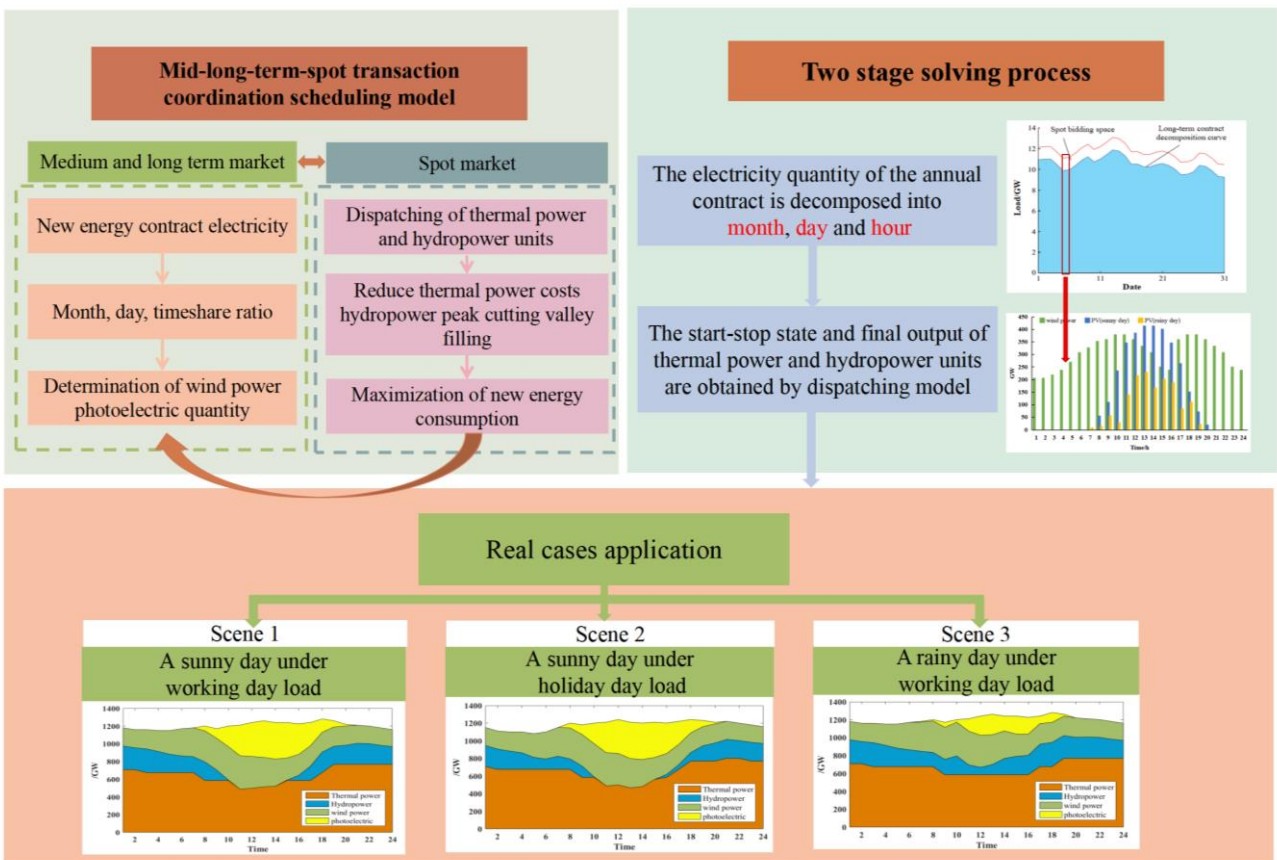

**Figure 3.** Schematic diagram of the MTCS coupled operation model.

### 3.2. Mid-Long-Term Spot Transaction Coordination Scheduling Model

The objective function for the MTCS model is composed of the fuel cost and start-up cost of the thermal power plant. The model includes the constraints of the power balance constraint, output limit constraint, system spinning reserve constraint, minimum start–stop time constraint, climbing rate constraint, inventory water volume constraint, and so on. The mathematical notations used in the MTCS model are shown in Table 1.

**Table 1.** These are the parameters provided for the proposed model.

| Functions | Description |
|---|---|
| $S_i^t$ | Start-up cost of thermal power unit *I*. |
| $N_T$ | Total number of thermal power units |
| $T$ | Total scheduling time |
| $P_{Ti}^t$ | The output power of thermal power unit *i* at time *t*. [MW] |
| $P_L^t$ | The load demand at time *t*. [MW] |
| $P_{Ti}$ | The actual output power of thermal power unit *T*. [MW] |
| $P_{Hi}$ | The actual output power of hydropower unit *H*. [MW] |
| $\overline{R}_L^t, \underline{R}_L^t$ | The positive and negative spinning reserve capacity required by the grid load at time *t*. [MW] |
| $\overline{R}_w^t, \underline{R}_w^t$ | The positive and negative spinning reserve capacity required by the wind farm at time *t*. [MW] |
| $r_{\mathrm{di}}, r_{\mathrm{ui}}$ | The lower limit and upper limit of the climbing rate of thermal power unit *i*. [MW/h] |
| $V_h^t$ | The storage volume of hydropower station *h* at time *t*. [$10^3$ m$^3$] |
| $\beta_1, \beta_2, \beta_3$ | The water consumption function coefficient of the output of the hydropower station unit and the quoted flow rate of the unit. |
| $P_{Hh}^t$ | The output power of hydropower station *h* at time *t*. |
| $H_h^t$ | The reservoir head of hydropower station *h* at time *t*. |

### 3.2.1. Objective Function

In terms of operating costs, the operating costs of thermal power plants are mainly fuel costs; while hydropower plants and other operating costs do not need to be paid, and the operating costs of hydropower, wind power, and photovoltaics are generally ignored. In order to minimize the fuel use of thermal power, the objective function consists of the fuel cost and start-up cost of the thermal power plant, as shown in the following formula.

$$\min F\left(P_{Ti}^t, U_i^t\right) = \sum_{t=1}^{T} \sum_{t=1}^{N_T} \left[ F\left(P_{Ti}^t\right) + S_i^t\left(1 - U_i^{t-1}\right) \right] \times U_i^t \tag{1}$$

where $F\left(P_{Ti}^t, U_i^t\right)$ is the power generation fuel cost function of the thermal power unit; $S_i^t$ is the start-up cost of thermal power unit *i* at time *t*; $U_i^t$ is the operating status of thermal power unit *i* at time *t*, where 1 means start, and 0 means stop; $N_T$ is the whole number of thermal power units; *T* means whole scheduling time; and $P_{Ti}^t$ is the output power of thermal power unit *I* at time *t*.

### 3.2.2. Constraint Condition

1. Power balance constraint:

$$\sum_{i=1}^{N_t} P_{Ti}^t + \sum_{j=1}^{N_H} P_{Hh}^t + \sum_{k=1}^{N_w} P_{Wk}^t + \sum_{p=1}^{N_p} P_{Pm}^t = P_L^t \tag{2}$$

where $P_{Hh}^t$ refers to the output power of hydropower unit *H* at time *t*; $P_{Wk}^t$ refers to the output power of wind farm *K* at time *t*; $P_{Pm}^t$ refers to the output power of photovoltaic farm *M* at time *t*; $N_H$ represents the whole number of hydropower units; $N_W$ represents the total number of wind farms; $N_P$ represents the total number of photovoltaic farms; and $P_L^t$ refers to the power grid at time *t* Load demand.

2. Output limit constraints:

$$
\begin{aligned}
P_{T\min i}U_i^t \leq P_{Ti}^t \leq P_{T\max i}U_i^t \\
P_{H\min i}U_h^t \leq P_{Hh}^t \leq P_{H\max i}U_h^t
\end{aligned}
\tag{3}
$$

where $P_{T\min i}$, $P_{H\min i}$ and $P_{T\max i}$, $P_{H\max i}$ are the minimum and maximum actual output power of the thermal power unit $t$ and the hydropower unit $h$, respectively.

3. System rotation and standby constraints:

$$
\begin{aligned}
\sum_{i=1}^{N_T}\left(P_{T\max i}-P_{Ti}^t\right)+\sum_{i=1}^{N_H}\left(P_{H\max h}-P_{Hh}^t\right) \geq \overline{R}_L^t + \overline{R}_W^t \\
\sum_{i=1}^{N_T}\left(P_{Ti}^t-P_{T\min i}\right)+\sum_{i=1}^{N_H}\left(P_{Hh}-P_{H\min h}^t\right) \geq \underline{R}_L^t + \underline{R}_W^t
\end{aligned}
\tag{4}
$$

where $\overline{R}_L^t$, $\underline{R}_L^t$ refers to the positive and negative spinning reserve capacity required by the grid load at time $t$; and $\overline{R}_w^t$, $\underline{R}_w^t$ refers to the positive and negative spinning reserve capacity required by the wind farm at time $t$.

4. Minimum start–stop time limit:

$$
U_i^t = \begin{cases}
1 & T_{\text{on } i}^{t-1} < T_{\text{up } i} \\
0 & T_{\text{off } i}^{t-1} < T_{\text{down } i} \\
0 \text{ or } 1 & \text{other}
\end{cases}
\tag{5}
$$

where $T_{\text{on } i}^{t-1}$ refers to the continuous online time of thermal power unit $i$ until $t-1$; $T_{\text{off } i}^{t-1}$ refers to the continuous offline time of thermal power unit $i$ until $t-1$; $T_{\text{up } i}$ is the minimum start − up time of thermal power unit $I$; and $T_{\text{down } i}$ is the minimum downtime of thermal power unit $i$.

5. Climbing rate constraints:

$$
-r_{\text{di}}U_i^t \leq P_{Ti}^t - P_{Ti}^{t-1} \leq r_{\text{ui}}U_i^t
\tag{6}
$$

where $r_{\text{di}}$, $r_{\text{ui}}$ refers to the lower limit and upper limit of the climbing rate of thermal power unit $i$.

6. Inventory water constraints:

$$
V_{\min h} \leq V_h^t \leq V_{\max h}
\tag{7}
$$

$$
V_h^t = V_h^0 + \sum_{t_c=1}^{t} q_h^{t_c} - \sum_{t_c=1}^{t-1} Q_h\left(P_{Hh}^{t_c}\right) - \sum_{t_c=1}^{t} d_h^{t_c}
\tag{8}
$$

where $V_h^t$ the storage volume of the hydropower station at the $t$; $V_{\min h}$, $V_{\max h}$ are the maximum storage volume and minimum storage volume of hydropower station $h$; $V_h^0$ is the stock water volume of hydropower station $h$ at the initial time of scheduling; $q_h^{t_c}$ is the inbound flow of the hydropower station $h$ at the moment $t_c$; $Q_h\left(P_{Hh}^{t_c}\right)$ is the quoted flow of the hydropower station $h$ at the moment $t_c$; and $d_h^{t_c}$ is the discarded water flow of the hydropower station $h$ at the moment $t_c$.

$$
Q_h^t = g\left(P_{Hh}^t \middle/ H_h^t\right) = \beta_0 + \beta_1 P_{Hh}^t + \beta_2\left(P_{Hh}^t\right)^2
\tag{9}
$$

where $\beta_1, \beta_2, \beta_3$ is the water consumption function coefficient of the output of the hydropower station unit and the quoted flow rate of the unit; $P_{Hh}^t$ is the output power of hydropower station $h$ at time $t$; $H_h^t$ represents the reservoir head of hydropower station $h$ at time $t$; and $\eta_h$ is the unit efficiency of hydropower station $h$.

### 3.3. Two-Stage Solution Process

　　The MTCS model is multi-agent, multi-period, and multi-energy, which can be determined by the joint dispatch of wind energy generation, PV power energy generation, thermal power, and hydropower. A two-stage solution method is introduced for this model, which includes electricity decomposition and unit start and stop status—adopting the contracted electricity decomposition method to decompose the medium- and long-term annual contract electricity. The dynamic programming (DP) method is then used to solve the optimal unit commitment involved in the MTCS model under the electricity decomposition results. The detailed solution process is described as follows (Figure 4).

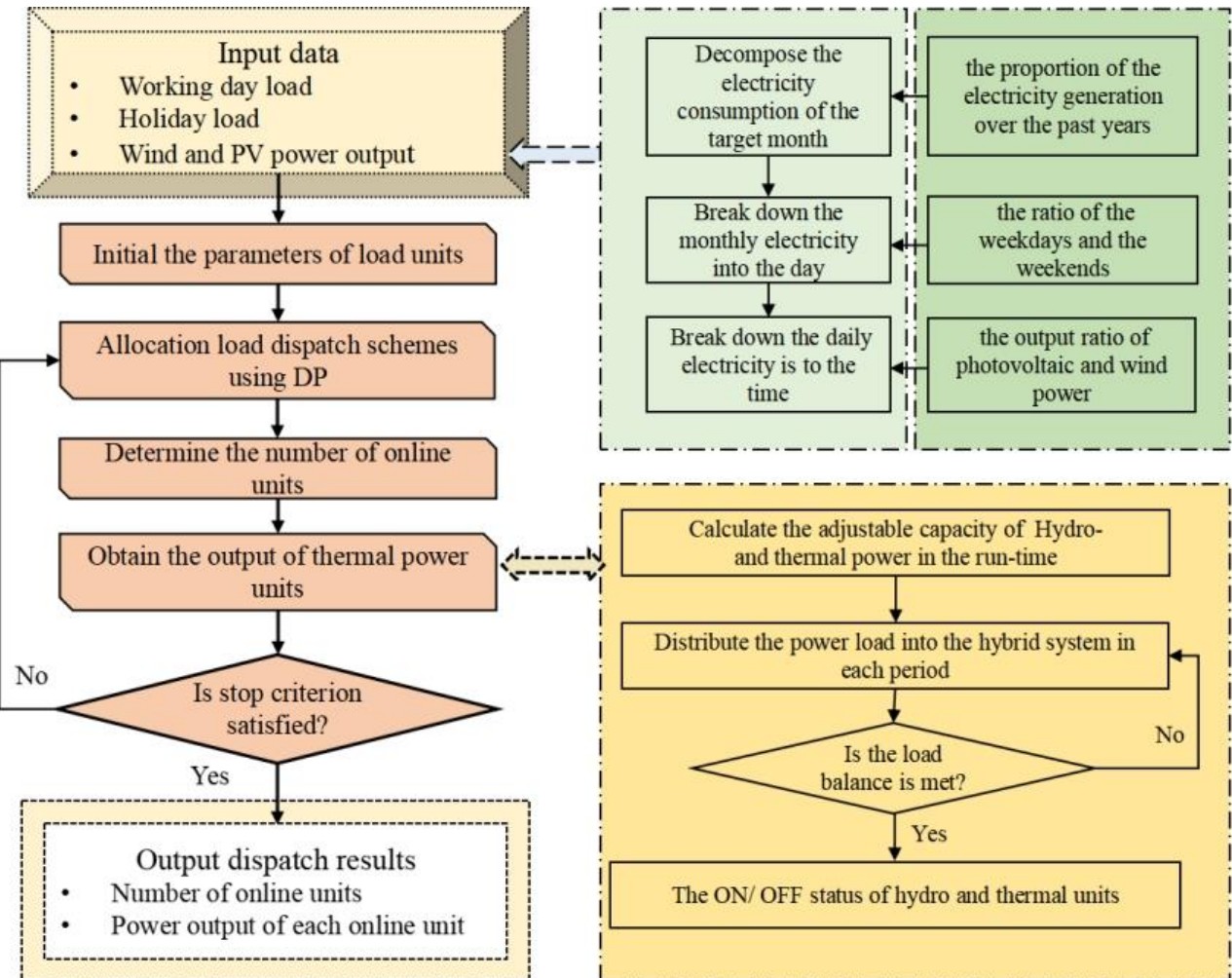

**Figure 4.** The flowchart of the proposed two-stage solution process.

### 3.3.1. Medium- and Long-Term Contract Electricity Breakdown

　　The medium- and long-term electricity contracts can maintain the high flexibility of the new energy medium- and long-term contract power and guide the hybrid energy system in the spot market. A medium- and long-term contract electricity decomposition method is established for dividing the multi-period electricity, which includes month, day, and hour. Determine the annual monthly electricity ratio based on the historical load adjustment, and then, set the weekday ratio $\alpha$, and the weekend ratio $\beta$ as the monthly daily ratio, selecting the contract electricity decomposition quantity $Q$ of a certain month in the contract for reference. The contract electricity decomposition situation is as follows.

$$Q_1 = \frac{Q}{\alpha \times 22 + \beta \times 8} \times \alpha \tag{10}$$

$$Q_2 = \frac{Q}{\alpha \times 22 + \beta \times 8} \times \beta \qquad (11)$$

where $Q$ is the contract quantity of electricity in a month; $Q_1$ is the daily power consumption on weekdays; and $Q_2$ is the daily power consumption on weekends, by million KWH.

### 3.3.2. Dynamic Programming

DP is an enumeration search method that can solve the optimization problem of the multi-stage decision-making process with a specific optimal proper [38]. Dynamic programming aims to find the best strategy to achieve the best results from all feasible decision sequences [39]. This paper decomposes the combined problem of water and thermal power into two sub-problems of combined hydropower and thermal energy. The peak load suppression method initially allocates the load borne by the hydropower and thermal energy. Then, the assigned quota value is solved by the integrated solution of the thermal power and hydropower. The detailed solving process is as follows:

1. Dividing the stages to decompose the problem into multiple interconnected steps appropriately in order to solve them in a particular order;
2. The state, which means the initial natural state or objective condition of each stage, is defined;
3. Making the decision, which means different choices that can be made when the process is in a particular state at a specific stage;

$$P_{k,n}(s_k) = \{u_k(s_k), u_{k+1}(s_{k+1}) \dots u_n(s_n)\} \qquad (12)$$

where $k$ is the stage variable, used to describe the stage variable, and $u_k(s_k)$ represents the decision variable when the state is in the $k$-th stage, which is a function of the state variable.

4. A strategy is formulated to a set of decisions arranged in order. The state transition equation refers to the evolution process of the decision process from one state to another, generally the evolution of two adjacent states; the index function is a quantitative indicator used to measure the quality of the process achieved.

$$V_{k,n} = V_{k,n}(s_k, u_k, \dots s_n, u_n, s_{n+1}) \qquad k = 0, 1, 2, \dots, n \qquad (13)$$

The optimal value of the index function in the dynamic programming method is called the optimal value function, which shows the process from the state of the $k$th stage to the end state of the $n$-th stage, the value of the index function achieved by the optimal strategy.

## 4. Case Analysis and Main Results

To verify the effectiveness of the multi-source complimentary dispatching model proposed in this paper to eliminate the phenomenon of abandoning wind and abandoning solar energy, we selected a case study that could minimize the problem size while still drawing representative results. Therefore, some standard units were chosen to form a microsystem to verify the model's applicability, and then different scenes were built to investigate the consequences of the MTCS model. This strategy takes the wind farms and the photovoltaic farms of Gansu New Energy Base, plus the thermal power stations and hydropower stations, as examples for the simulation analysis.

### 4.1. Energy Type and Unit Parameter Setting

The units include four thermal power plants with a total of six thermal power units, two hydropower stations, one wind farm, and one photovoltaic power station. The installed capacity of the wind farm is 500 MW; the installed capacity of the PV farm is 450 MW, and the other specific parameters are shown in Table 2 (T1–T6 are thermal power units, H1–H2 are hydropower stations). Scene 1 expresses a sunny day under a typical working-day load;

scene 2 represents a sunny day under a holiday load; and scene 3 is a rainy day under a standard active-day load.

**Table 2.** Thermal power and hydropower unit parameters.

| Unit | T1 | T2 | T3 | T4 | T5 | T6 | H1 | H2 |
|------|----|----|----|----|----|----|----|----|
| $P_{i,\min}$/MW | 250 | 250 | 110 | 110 | 100 | 100 | 0 | 0 |
| $P_{i,\max}$/MW | 600 | 600 | 330 | 330 | 300 | 300 | 225 | 30.4 |

*4.2. Electric Quantity Decomposition Result*

The power generation determines the monthly ratio of new energy power generation in historical years. Then, the power is decomposed into the weekdays and weekends by setting the factors, such as α β. Finally, the output ratio of the photovoltaic and the wind power in this area is divided into the power generation of each time; this realizes the monthly, daily, and hourly decomposition of the medium- and long-term electricity. According to the data of the selected historical years, the power generation ratio of each energy in each month is determined in order to determine the power of each energy unit in the target month; then, via Equations (10) and (11), the monthly power is calculated by day according to the difference between the working days and the holidays. According to the output curves of the photovoltaic and the wind power in different time periods under different weather conditions, the daily power consumption is calculated by time sharing. The detailed process is shown in Figure 5.

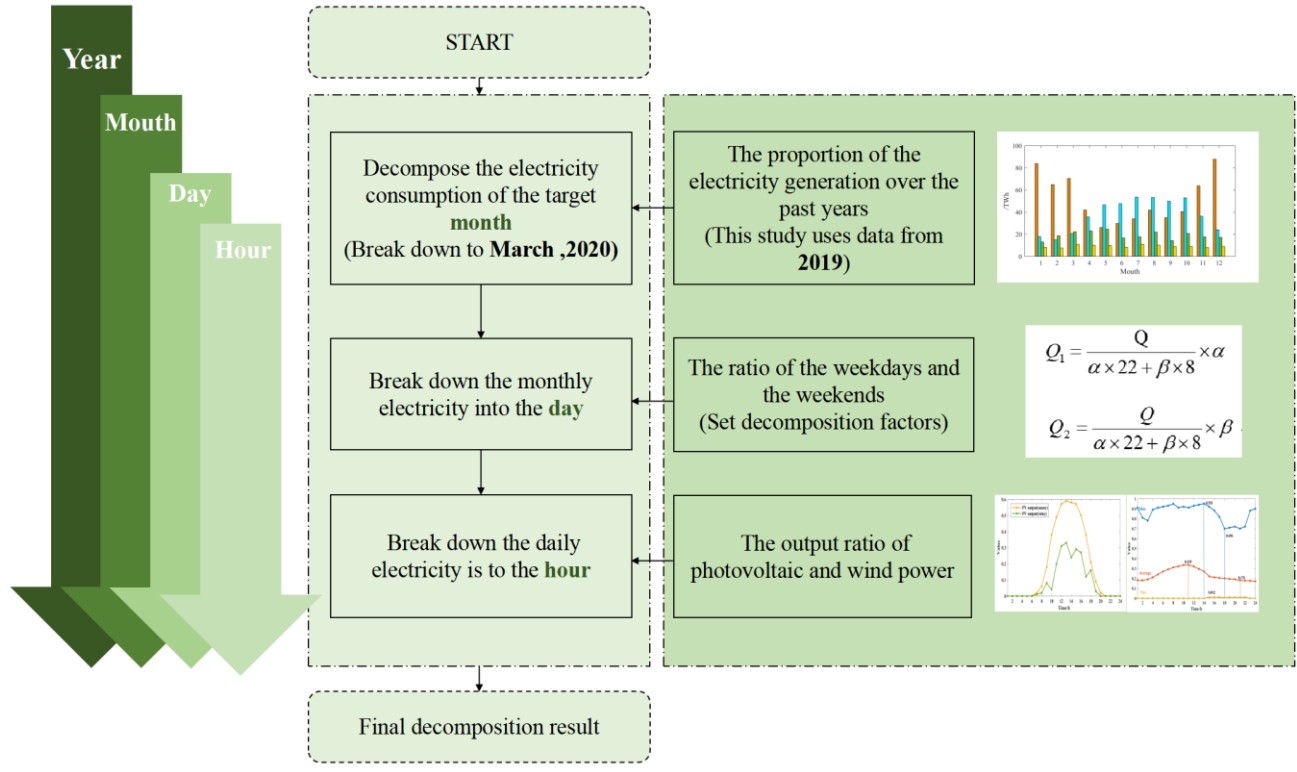

**Figure 5.** Electricity decomposition process.

Using the power generation in Gansu Province in 2019 as the historical data, the target year's monthly power share is determined, as shown in Figure 6. The analysis of Figure 6 shows that the power generation of various energy sources has seasonality. Due to the constraints of ensuring heating and power grid security, the thermal power output is concentrated in the heating season from November to March of the following year, and the annual power generation presents a "concave" curve. Hydropower is abundant in flood

seasons but is affected by the decrease in incoming water and the regulation of the Yellow River group. The annual power generation shows a convex curve. New energy generation equipment is highly uncertain and subject to weather conditions. The comparison between March and August in 2019 shows that the proportion of thermal power generation is the highest in March and that of the hydroelectric power generation is in August. This result also fully proves the seasonality of the energy generation. Natural conditions limit renewable energy sources. It is known that electricity consumption in March 2020 was 10.359 TWh, which will be used as the basis of the electricity decomposition. According to the historical load situation in 2019, the monthly ratio is determined, and the contract is broken down by monthly electricity. The results are shown in Table 3.

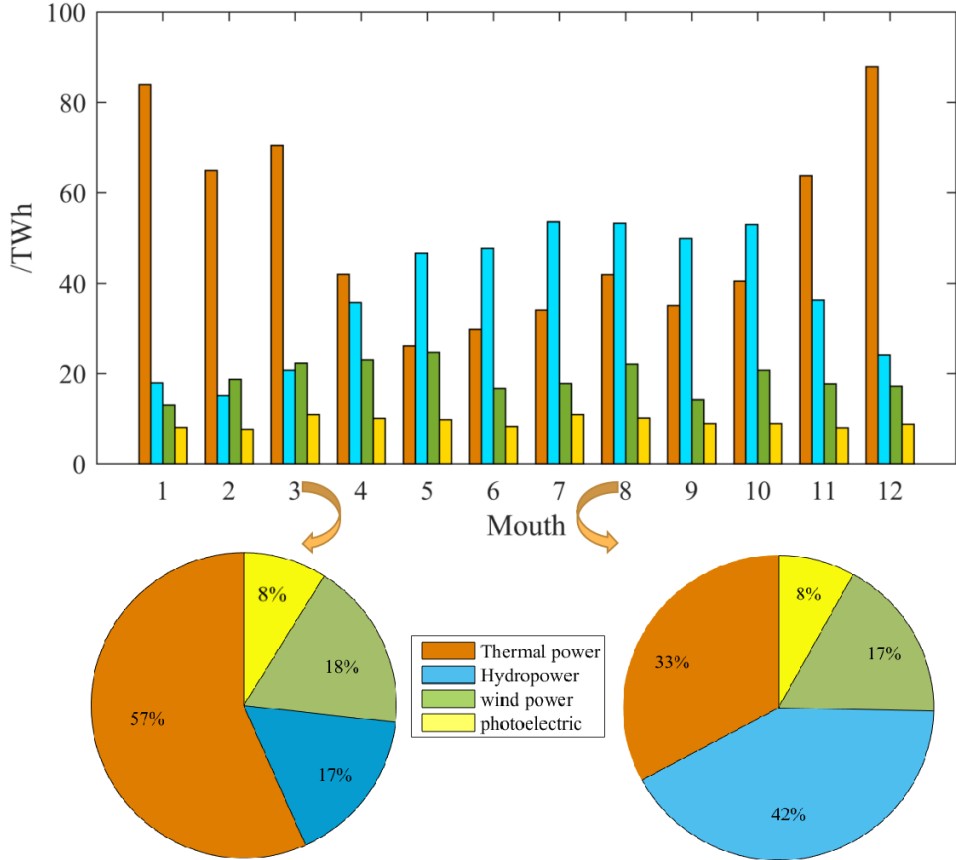

**Figure 6.** Schematic diagram of the proportion of electricity generation in Gansu Province in each month of 2019.

**Table 3.** Monthly electricity/TWh.

| Type of Power Supply | Should Generate Electricity | Proportion |
| --- | --- | --- |
| Thermal power | 58.65 | 56.62% |
| Hydropower | 17.22 | 16.63% |
| Wind power | 18.56 | 17.91% |
| Photoelectric | 9.16 | 8.84% |

The daily power ratio was set to a percentage of $\alpha$, taken as 1 for the weekdays, and $\beta$, as 0.85 for the weekends. The daily breakdown of contract electricity is shown in Table 4. After the daily power is determined, the output ratio of each period is determined according to the average output of the PV (sunny, cloudy, and rainy days) and wind power in Gansu Province, as shown in Figures 7 and 8, to determine the periods of wind power

and PV power generation in the daily power generation. The results are shown in Table 5 and Figure 9.

**Table 4.** Electricity per day (/TWh).

| Type of Power Supply | Workdays Power Generation | Weekends Power Generation |
|---|---|---|
| Thermal power | 2.03 | 1.73 |
| Hydropower | 0.61 | 0.52 |
| Wind Power | 0.64 | 0.55 |
| Photoelectric | 0.32 | 0.27 |

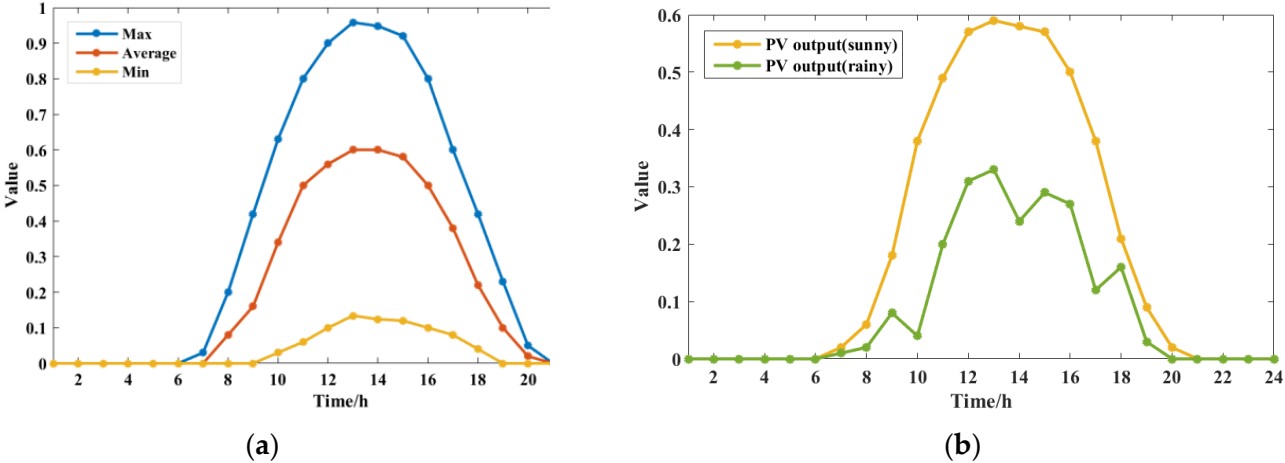

**(a)**                                                            **(b)**

**Figure 7.** Statistics of photovoltaic output by periods in Gansu Province: (**a**) average, maximum, and minimum output; (**b**) photovoltaic output on sunny, cloudy, and rainy days.

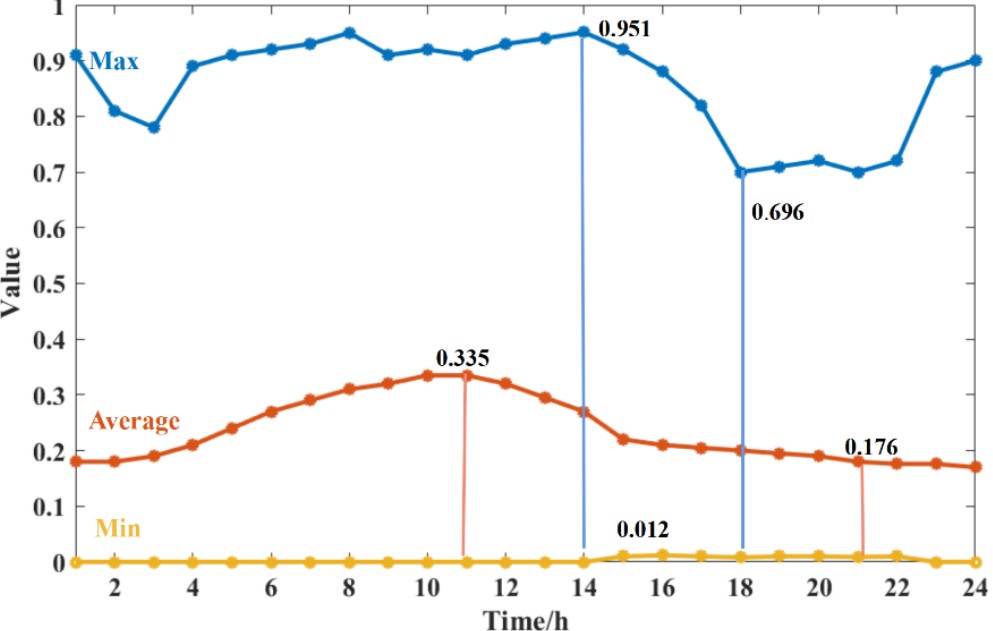

**Figure 8.** Statistics of wind power processing by time in Gansu Province.

**Table 5.** Time of period of photovoltaic and wind power (GW).

| Period | Wind Power | PV Power | |
| --- | --- | --- | --- |
| | | Sunny | Rainy |
| 1 | 204.8 | 0 | 0 |
| 2 | 204.8 | 0 | 0 |
| 3 | 217.6 | 0 | 0 |
| 4 | 236.8 | 0 | 0 |
| 5 | 268.8 | 0 | 0 |
| 6 | 307.2 | 0 | 0 |
| 7 | 326.4 | 0 | 6.98 |
| 8 | 352 | 54.4 | 13.8 |
| 9 | 358.4 | 108.8 | 55.4 |
| 10 | 377.6 | 233.6 | 27.6 |
| 11 | 377.6 | 345.6 | 138.4 |
| 12 | 358.4 | 384 | 214.6 |
| 13 | 332.8 | 412.8 | 228.4 |
| 14 | 307.2 | 412.8 | 166.2 |
| 15 | 249.6 | 400 | 200.8 |
| 16 | 236.8 | 345.6 | 186.9 |
| 17 | 230.4 | 262.8 | 83.1 |
| 18 | 224 | 150.4 | 110.8 |
| 19 | 217.6 | 70.4 | 20.8 |
| 20 | 217.6 | 19.2 | 0 |
| 21 | 204.8 | 0 | 0 |
| 22 | 198.4 | 0 | 0 |
| 23 | 198.4 | 0 | 0 |
| 24 | 192 | 0 | 0 |

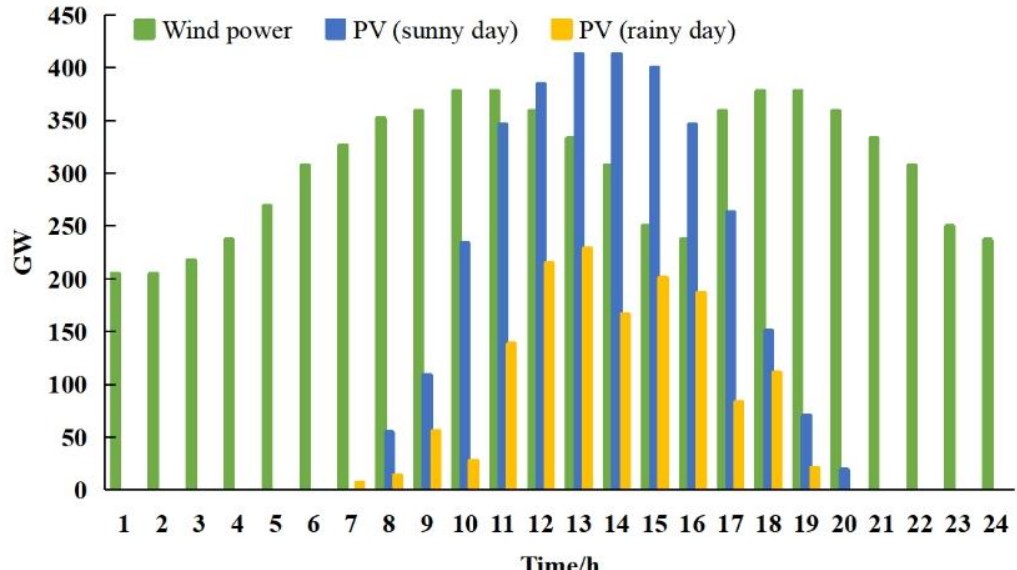

**Figure 9.** Output of wind power and PV power.

*4.3. Scheduling Model Results*

According to the two-stage solution process, the sequence combination is optimized and modified during the entire operation period: 1 means start, and 0 means shutdown. Finally, the start and stop states and the output of the hydropower and thermal power units in the three scenes are obtained, as shown in Figure 10. The scheduling costs of the three scenarios are counted, and the cost comparison results are shown in Table 6.

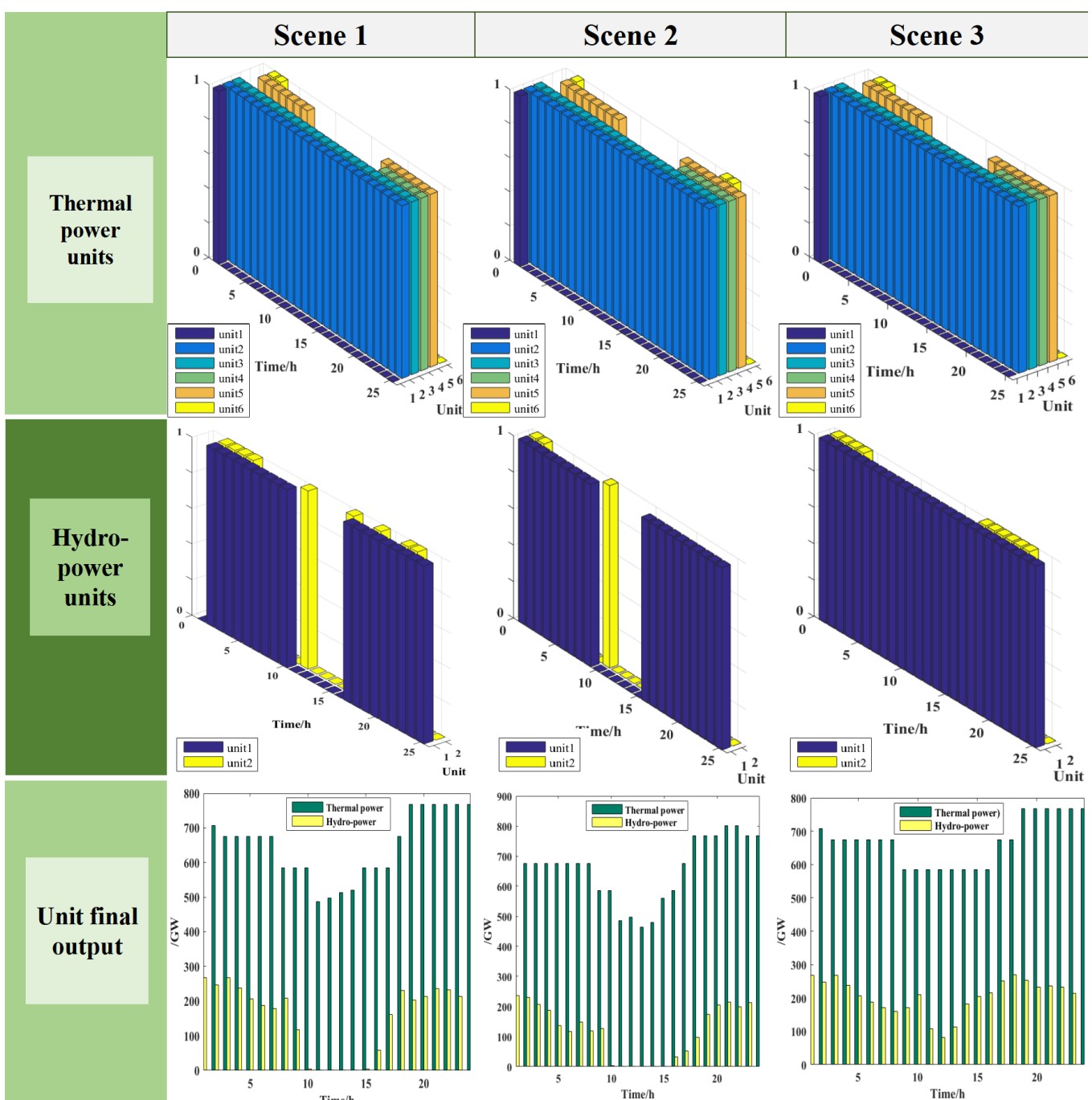

**Figure 10.** The start and stop states and output for thermal power units and hydropower units in the three scenes.

**Table 6.** Scheduling cost comparison.

|  | Start–Stop Costs/RMB | Running Costs/RMB | Total Cost/RMB |
|---|---|---|---|
| Scene 1 | 2830 | 229,893 | 232,723 |
| Scene 2 | 2860 | 224,536 | 227,396 |
| Scene 3 | 2830 | 248,578 | 251,408 |

Scene 1 represents a sunny day under a typical working-day load; due to the priority consumption of the wind power and photovoltaic power, only T2 and T3 of the thermal

power units are activated from period 8 to period 17; starting from period 18, due to the apparent reduction in the photovoltaic power output, the thermal power units increase and T3 and T4 are put into operation; from periods 1 to 10 and periods 15 to 24, the hydropower units are put into operation. Scene 2 represents a sunny day under a holiday load. From periods 9 to 16, only T2 and T3 of the thermal power units are activated. From period 21–22, due to the apparent reduction in the photovoltaic output and the peak period of electricity consumption during the holidays, the thermal power and the units T3 and T4 were added to the team and put into operation. The hydropower units were put into operation during periods 1 to 9 and 15 to 24. Scene 3 represents a rainy day under a typical working-day load. From period 9 to period 16, thermal power units T2 and T3 are in the starting state; due to the rainy weather, the photovoltaic output is significantly reduced, compared to the periods 11 to 14, when the photovoltaic output peaks in scenes 1 and 2 are put into the hydropower unit for peak shaving.

In the three scenes, thermal power units 2 and 3 are normally kept open and bear the baseload. The difference lies in the opening period of units 3 and 6. In addition, the hydropower unit H1 remains on in scene 3 to fill up the lack of photovoltaic output.

The analysis of Table 6 shows that the start–stop costs of the three scenarios are not much different, and the difference is mainly in the operating costs. Scene 2 increased by RMB 5357 compared with scene 1, and scene 3 increased by RMB 18,685 compared with scene 1. By comparison, it was found that the reason for the difference between scene 2 and scene 1 is the gap between the load on holidays and the load on working days. In addition, the reason for the gap between scene 3 and scene 1 is that the photovoltaic output drops significantly on rainy days, and the hydropower and thermal power need to be dispatched to participate in the operation, resulting in increased costs.

## 5. Discussion

The MTCS model is determined by the joint dispatch of wind energy generation, PV power energy generation, thermal power, and hydropower. A two-stage solution method to solve this model includes electricity decomposition and unit start and stop status. The contract power decomposition method decomposes the medium- and long-term annual contract power, yielding the output of new energy sources (Figure 11). The dynamic planning (DP) method is then used to obtain the outcome of the thermal and hydro units (Figure 12). This section discusses the results of the three scenarios mentioned above, including two relevant and essential aspects.

They compared the output results of scenes 1 (under the work-day load) and 2 (under the holiday load) under the same sunny conditions but with different loads. Regarding the thermal power output, the peak power generation period of scene 1 is from 19:00 to 24:00, and scene 2 is from 21:00 to 22:00. The reason is that the photovoltaic output is significantly reduced at this time. Regarding the production of the hydropower units, the same thing is that during the period when the photovoltaic power output is abundant, the hydropower units are out of operation from the hours of 10:00–15:00, and the peak times of production are both at 1 o'clock.

Under the same load but different weather conditions, scene 1 is sunny, and 3 is rainy. The significantly different photovoltaic output will affect the output result of this hybrid system. Regarding the thermal power output, scene 3 has two periods more than scene 1 to open one more unit. The peak power generation period of scene 3 is also from 19:00 to 24:00. Still, due to the periods 11–13, when the photovoltaic output is significantly reduced, the thermal power output has increased substantially, as shown in Figure 12d. Regarding the production of the hydropower unit, unit H1 of scene 3 typically remains open for six more periods than in scene 1.

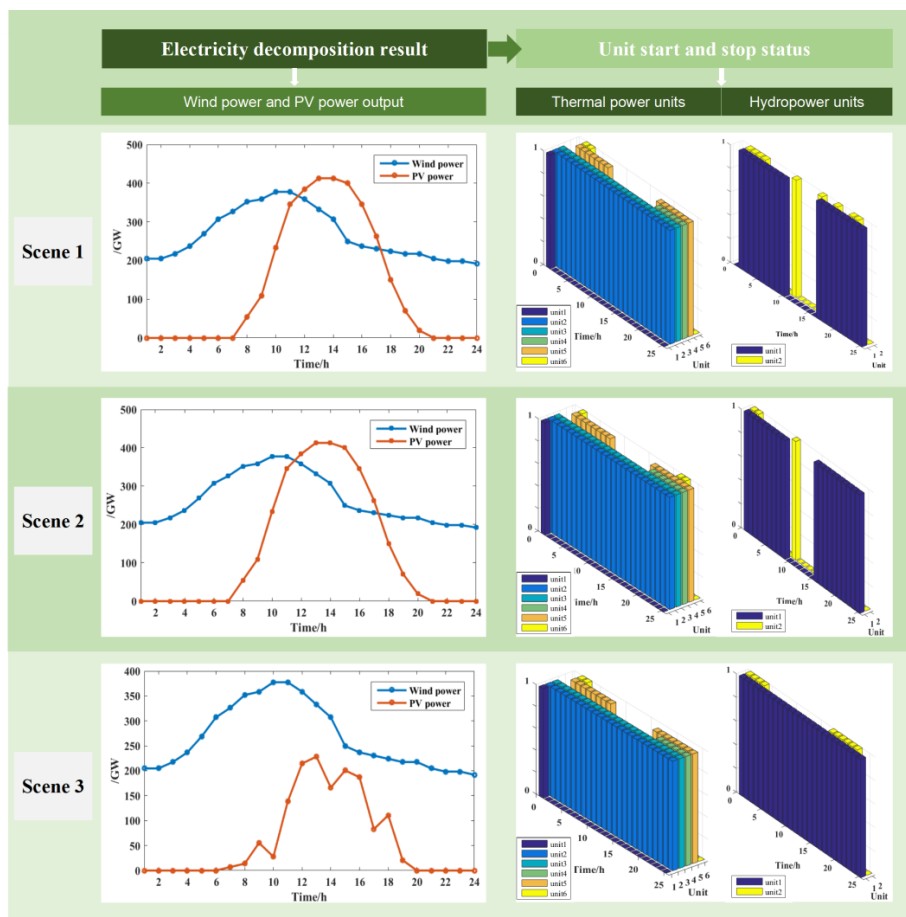

**Figure 11.** The start and stop results of the unit and the output of wind power and photovoltaic.

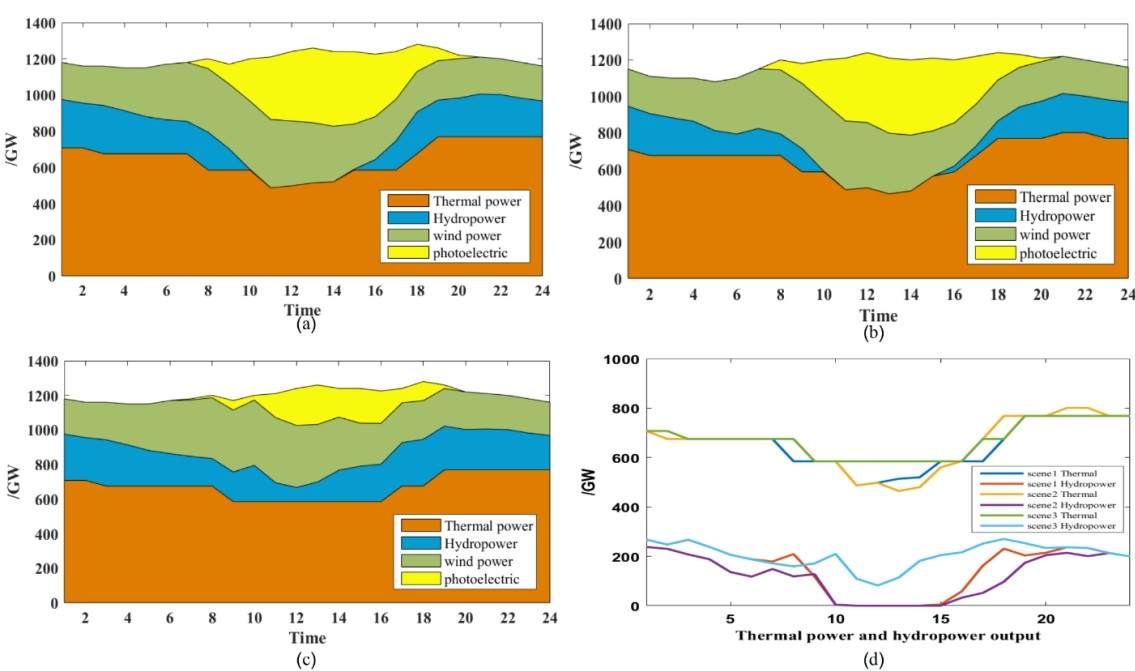

**Figure 12.** The output result of simulation scenes 1 (**a**), 2 (**b**), and 3 (**c**); the power output of thermal power and hydropower units is depicted in (**d**).

Through the experimental results of the three scenes from the application of the Gansu electric power field pilot, the MTCS model can not only keep the new energy unit stable but can help to promote the ability to absorb new energy power. Specifically, the increased external load is mainly borne by the thermal power unit, followed by the hydropower unit in this integrated energy system. Thermal power generating units can take the primary load, and the hydropower generating units can cut peaks and fill valleys to ensure that the new energy sources are preferentially consumed. The results show that the model can facilitate the consumption of new energy based on the concept of maintaining the stable operation of new energy units and achieving complete absorption in market transactions and can be popularized and used. Hence, with the new energy installed capacity expansion, the thermal power units will gradually degrade.

## 6. Conclusions

Due to the intermittent and anti-peak shaving characteristics of the new energy generator sets, the phenomenon of large-scale power absorption and power abandonment will be intensified. The centralized market was produced to resolve the problem of power absorption and power abandonment, which can use medium- and long-term transactions to make a contract for differences and then cooperate with the spot market to manage the risks of the electricity market. Although the policy documents supporting the acceleration of China's power market reform process have been improved, the action of multi-energy coordinated power market transactions has not been implemented due to the lack of a specific spot market mechanism. Therefore, the concept of the joint operation of the multi-energy hybrid electricity market and the coupled trading of multi-energy complementary systems is introduced, which can take advantage of the complementarity and substitution between the different energy sources to achieve flexibility in energy production, consumption, storage, and transmission, optimize resource allocation, and absorb renewable energy on a larger scale.

According to the coordinated operation characteristics of the centralized power spot market in pilot areas of China, this study analyzes the operation situation of the power spot market, uncovers the problems existing in the operation process of the spot market, and then proposes an MTCS model for the multi-energy system by considering the medium- and long-term electricity market uncertainty and the trial operation characteristics of the spot power market of China. The results of testing this model on the Gansu region, one of the first eight spot pilot areas in China, have been presented and discussed. Compared with models that only consider medium- and long-term power market transactions, the MTCS model can identify the operational uncertainties brought by the opening of the spot to the medium- and long-term trading system, fundamentally promoting the development and consumption and ensuring the utilization of new energy. Additionally, applying this model to scene 1 (typical work-day load on a sunny day), scene 2 (bright holiday load), and scene 3 (rainy-day load on a cloudy day) can obtain the results that the thermal power units are responsible for the increase and decrease in the pack and that the new energy units are maintaining a relatively stable operation. In summary, the proposed MTCS model can promote the efficient participation of new energy generation units in the spot market and ensure the priority of new energy generation in the spot market. It has also been proved that the coordinated operation of the energy system can make full use of the complementary and alternative characteristics of different energy sources in time and space, compensate for the uncertainty of renewable energy, and promote the total consumption of renewable fuel.

This research fills in the blanks of the theoretical and application guidance for coordinating medium- and long-term transactions in spot transactions with multi-energy systems in the context of large-scale new energy participation. Future work in this area will mainly focus on the multi-energy coupling operation, an integrated energy system participating in the spot market transaction, which can fundamentally solve the problem of new energy consumption. Specifically, the research group will construct the dynamic model of the multi-energy complementary power market driven by multi-scale prediction, develop an

efficient and intelligent multi-energy complementary management and decision-making system, and so on.

**Author Contributions:** Conceptualization, X.W. and K.W.; methodology, X.W.; software, X.W. and Y.L.; validation, R.J. and J.D.; formal analysis, H.D.; writing—original draft preparation, X.W.; writing—review and editing, K.W.; visualization, Y.L. All authors have read and agreed to the published version of the manuscript.

**Funding:** This research was funded by Shaanxi Province Science and Technology Project (2022JM-208) and the Key Industry Innovation Chain Project of the Science and Technology Department of Shaanxi Province (2019ZDLGY18-03).

**Institutional Review Board Statement:** Not applicable.

**Informed Consent Statement:** Not applicable.

**Data Availability Statement:** The data presented in this study are available on request from the corresponding author.

**Acknowledgments:** The author is grateful to the editors and anonymous referees for their comments and suggestions.

**Conflicts of Interest:** The authors declare no conflict of interest. The funders had no role in the design of the study; in the collection, analyses, or interpretation of data; in the writing of the manuscript; or in the decision to publish the results.

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
