# Peer review of "Research on Coupled Cooperative Operation of Medium- and Long-Term and Spot Electricity Transaction for Multi-Energy System: A Case Study in China"

_sustainability, doi:10.3390/su141710473_

Round 1

Reviewer 1 Report

This paper puts forward the model of cooperative medium and long term and spot trading, which can promote the consumption of new energy to a certain extent.The study is interesting and valuable, while few issues need further clarification in line with the following comments.

 1.   What are the advantages of the proposed two-stage method to solve the collaborative operation model compared with the existing methods?

2.   Please pay attention to the uniform format of the table, such as the thickness of the box line.

3.   The title of figure 10 is too complicated. Please be precise and brief.

4.   The manuscript is too long, please remove unnecessary descriptions.

Reviewer 2 Report

- The manuscript "Research on Coupled Cooperative Operation of Medium-Long-term and Spot Electricity Transaction for Multi-energy system A Case Study in China"  merits a high rating for the subject's specialization.

- The methodological approach meets the minimum requirements commonly specified, and the results are of particular scientific interest.

- If the authors see fit for the discussion section, please debate the results and explain how they might be interpreted in light of the previous studies.

Reviewer 3 Report

Dear authors,

 I appreciate your support by selecting the Sustainability for possible publication of your research work.

The paper develops a mid-long-term spot transaction coordination scheduling model for a multi-energy  system  taking into account the medium long-term electricity market uncertainty and the trail operation characteristics of the spot power market, the model being tested on the Gansu region, from China.

The paper is well structured, combining a comprehensive literature research with the mathematical models, which led to valuable results, which have to be published

However, the paper has a few points which make me recommend a minor revision of the manuscript:

·       The schematic diagram of the MTCS coupled operation model depshown in Fig.3 have to be explained.

·       The electric quantity decomposition process depicted in fig. 5 also need to be explaind in text.

It is a fair research article submission and in my opinion just these few minor revisions should be addressed before publication.

Sincerely yours,
